

# Skin bacterial community differences among three species of co-occurring Ranid frogs

Zachary Gajewski[1], Leah R. Johnson[1,2], Daniel Medina[1],
William W. Crainer[3], Christopher M. Nagy[4] and Lisa K. Belden[1]

[1] Department of Biological Sciences, Virginia Polytechnic Institute and State University (Virginia Tech), Blacksburg, Virginia, United States
[2] Department of Statistics, Virginia Polytechnic Institute and State University (Virginia Tech), Blacksburg, Virginia, United States
[3] Department of Animal Sciences, Cornell University, Ithaca, New York, United States
[4] Mianus River Gorge Preserve, Bedford, New York, United States

Corresponding author
Zachary Gajewski, gzach93@vt.edu

## ABSTRACT

Skin microbial communities are an essential part of host health and can play a role in mitigating disease. Host and environmental factors can shape and alter these microbial communities and, therefore, we need to understand to what extent these factors influence microbial communities and how this can impact disease dynamics. Microbial communities have been studied in amphibian systems due to skin microbial communities providing some resistance to the amphibian chytrid fungus, *Batrachochytrium dendrobatidis*. However, we are only starting to understand how host and environmental factors shape these communities for amphibians. In this study, we examined whether amphibian skin bacterial communities differ among host species, host infection status, host developmental stage, and host habitat. We collected skin swabs from tadpoles and adults of three Ranid frog species (*Lithobates* spp.) at the Mianus River Gorge Preserve in Bedford, New York, USA, and used 16S rRNA gene amplicon sequencing to determine bacterial community composition. Our analysis suggests amphibian skin bacterial communities change across host developmental stages, as has been documented previously. Additionally, we found that skin bacterial communities differed among Ranid species, with skin communities on the host species captured in streams or bogs differing from the communities of the species captured on land. Thus, habitat use of different species may drive differences in host-associated microbial communities for closely-related host species.

## INTRODUCTION

Bacteria that live on the skin of host are dynamics communities that can be shaped by changing environmental and host factors; such as host species, host lifestyle, season, pH, and temperature (*Moitinho-Silva et al., 2021*; *Rebollar, Martínez-Ugalde & Orta, 2020*; *Jiménez & Sommer, 2017*; *Varela et al., 2018*; *McKenzie et al., 2012*). These bacterial

communities can serve a variety of important functions, including influencing host health (*Rebollar, Martínez-Ugalde & Orta, 2020*; *Jiménez & Sommer, 2017*; *Cho & Blaser, 2012*; *Trevelline et al., 2019*). Due to their health benefits, skin microbial communities have been studied in the context of several wildlife disease systems, and in some cases these communities can mitigate the effects of diseases (*Ritchie, 2006*; *Hoyt et al., 2015*; *Hoyt et al., 2019*; *de Bruijn et al., 2018*; *Yang et al., 2017*; *Harris et al., 2009b*). Using manipulation of microbial communities to manage wildlife diseases has been suggested, but a better understanding of what factors shape these communities is needed first to ensure maximal effectiveness of interventions (*Rebollar, Martínez-Ugalde & Orta, 2020*; *McKenzie, Kueneman & Harris, 2018*; *Bletz et al., 2013*).

The amphibian chytrid fungus is an excellent model to study the relationship between disease and host skin microbial communities (*Rebollar, Martínez-Ugalde & Orta, 2020*; *Burkart et al., 2017*; *Harris et al., 2009a*). The fungal pathogen that causes chytridiomycosis, *Batrachochytrium dendrobatidis* (Bd), has been linked to amphibian population declining globally (*Scheele et al., 2019*; *Fisher, Garner & Walker, 2009*; *Stuart et al., 2004*; *Berger et al., 1998*). Naturally occurring bacteria found on the skin of amphibians, such as *Janthinobacterium lividum*, can offer some resistance to Bd through the production of anti-fungal metabolites (*Harris et al., 2009a*, *2009b*; *Becker et al., 2009*). Additionally, differences in skin bacterial communities between amphibians have been associated with differences in infection prevalence. For example, populations with more Bd inhibitory bacteria have lower Bd infection (*Burkart et al., 2017*; *Lam et al., 2010*; *Woodhams et al., 2007*). These relationships between the amphibian skin microbiome and severity of chytridiomycosis has motivated a number of studies examining host, environmental, and pathogen factors that shape the skin microbial communities (*Jani et al., 2021*; *Rebollar, Martínez-Ugalde & Orta, 2020*; *Abarca et al., 2018*; *Belden et al., 2015*; *Estrada et al., 2019*, *Varela et al., 2018*). However, not all factors are fully understood and by sampling wild amphibian populations we can continue monitoring Bd prevalence and examine host bacterial communities and factors that drive variation across environments and hosts.

In this study, we assessed Bd infection and the skin microbiome in natural, un-manipulated amphibian populations of amphibians at the Mianus River Gorge Preserve (MRGP) in Bedford, New York, USA. We collected skin bacterial community data from adults of three Ranid frog species (*Lithobates clamitans*, *Lithobates sylvaticus*, and *Lithobates palustris*) and from *Lithobates* tadpoles, to determine how bacterial communities might differ among Ranid species and across life stages. For all the species listed above, and for *Lithobates catesbeianus*, we also collected Bd infection data. Due to previous studies finding low infection prevalence in the Northeastern United States (*Richards-Hrdlicka, Richardson & Mohabir, 2013*), we expected infection rates to be low and uniform across the sampled amphibian species. Based on previous work, we also expected host species to drive differences in skin bacterial communities rather than sampling locations, infected individuals to have different skin bacterial communities, and that tadpole bacterial communities would differ from those of adults (*Abarca et al., 2018*;

**Table 1 The number of amphibians sampled, infected, and sent for Illumina sequencing.** Number of individuals sampled at each sites and the number of individuals infected at the site by species. Sample sizes for 16S rRNA gene amplicon sequencing are shown in parentheses in bold. Species were sampled at three field sites at Mianus River Gorge Preserve, Bedford, New York, USA.

| Sites | Amphibian swabs | Infected | | Not infected | | Total caught | |
|---|---|---|---|---|---|---|---|
| | | 2017 | 2018 | 2017 | 2018 | 2017 | 2018 |
| Site 1 | *L. sylvaticus* | 4 (**4**) | 1 | 20 (**13**) | 27 | 24 (**17**) | 28 |
| | *L. clamitans* | 0 | 0 | 19 (**12**) | 27 | 19 (**12**) | 27 |
| | *L. catesbeianus* | 0 | 0 | 0 | 0 | 0 | 0 |
| | *L. palustris* | 12 (**12**) | 16 | 13 (**13**) | 11 | 25 (**25**) | 27 |
| | *L.* Tadpoles | 1 | 0 | 39 (**12**) | 22 | 40 (**12**) | 22 |
| Site 2 | *L. sylvaticus* | 2 | X | 24 | X | 26 | X |
| | *L. clamitans* | 2 | X | 20 | X | 22 | X |
| | *L. catesbeianus* | 1 | X | 11 | X | 12 | X |
| | *L. palustris* | 0 | X | 0 | X | 0 | X |
| | *L.* Tadpoles | 13 | X | 19 | X | 32 | X |
| Site 3 | *L. sylvaticus* | X | 1 | X | 25 | X | 26 |
| | *L. clamitans* | X | 0 | X | 26 | X | 26 |
| | *L. catesbeianus* | X | 0 | X | 0 | X | 0 |
| | *L. palustris* | X | 5 | X | 25 | X | 30 |
| | *L.* Tadpoles | X | 0 | X | 22 | X | 22 |
| Total | | 35 (**16**) | 23 | 165 (**50**) | 185 | 200 (**66**) | 208 |

*Belden et al., 2015*; *Kueneman et al., 2014*; *McKenzie et al., 2012*). Portions of this text were previously made available as part of a dissertation (*Gajewski, 2021*).

# METHODS

## Sample collection

### Field site

We captured, swabbed, and released amphibians at three different sites on or near the Mianus River Gorge in Bedford, New York, USA, to detect Bd infection and to collect samples of the skin bacterial communities (Table 1; Fig. 1). The sampling took place in June 2017 and 2018. Site 1 was located behind a residential communities, accessible by a dirt road. Site 1 consisted of a stream that varied in forest cover and the density of understory cover and split off into an open wetland area. Additionally, across an open meadow there was a vernal pool with partial forest cover. Site 1 was the only site where skin bacterial communities were examined, and therefore, was divided into five subsites based on differences in habitat (Fig. 1). Amphibians were sampled at site 1 in 2017 and 2018, but skin bacterial samples were only taken from 2017. Site 2 consisted of a roadside pond with no forest cover, on private land, that led to a wetland that in turn fed into another pond, both on MRGP property. The wetland and ponds on MRGP property both had partial forest cover and had a dense understory dominated by *Symplocarpus foetidus* (skunk

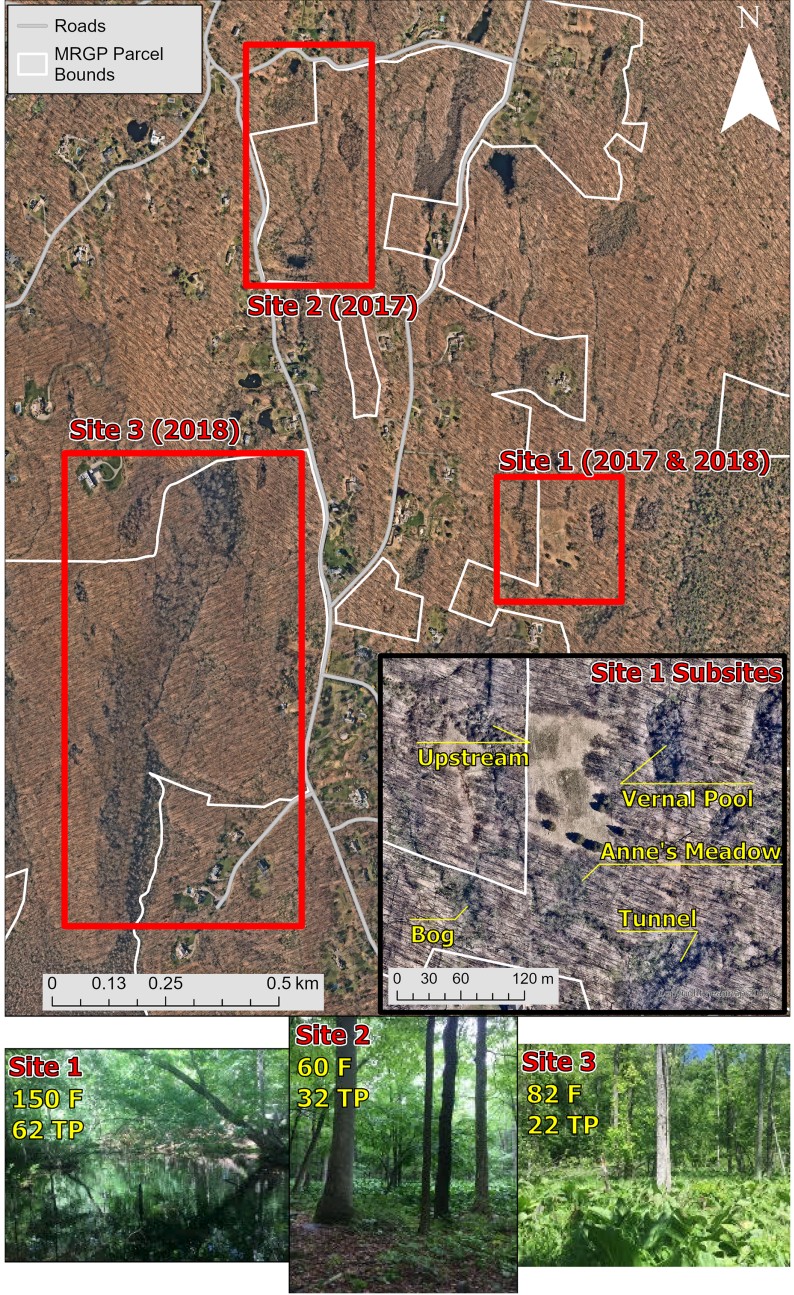

**Figure 1 Map and pictures of sampling locations.** The top map shows the main three sites sampled at the Mianus River Gorge Preserve. The smaller map at the bottom right of the top map, shows the subsites at site 1, where all the amphibian bacterial samples were collected from. The bottom row of pictures show sites where amphibians were samples from. Site 1 consisted of a vernal pool and small nearby streams, sampled in 2017 and 2018. Site 2 was a marshland covered mostly in skunk cabbage that had a pond on both sides, sampled just in 2017. Lastly, site 3 was a marshland that was fed by larger streams, sampled just in 2018. Aerial photographs copyrighted (*Nearmap US Vertical Imagery, 2015*).

cabbage). All three of these water bodies were sampled as a single site (site 2) in 2017. Site 3 was along a stream that started in a forested area near two vernal pools and led into a thick marshland with less forest cover. Amphibians were collected near the marshland and at the vernal pools in 2018.

### Amphibian sampling

We collected skin swabs from *L. sylvaticus*, *L. clamitans*, *L. catesbeianus*, and *L. palustris*. At all three sites, we collected samples from *L. sylvaticus* and *L. clamitans*, while *L. catesbeianus* was only sampled at Site 2, and *L. palustris* was only found and sampled at sites 1 and 3 (Table 1). We also sampled Lithobates tadpoles, which were not identified to species, at site 1 in 2017 and 2018, at site 2 in 2017, and at site 3 in 2018 (Table 1). The use of animals was approved by Virginia Tech's Institutional Animal Care and Use Committee (16-193-STAT) and the state of New York (permit #2213).

We sampled only one site and one species per day. Using dip nets, we caught amphibians, which were then placed individually into sterile Whirl-Pak bags until our sample size at the site was reached to avoid re-sampling individuals (adults per site: 20 in 2017 and 25 in 2018; tadpoles per site: 30 in 2017 and 20 in 2018). For tadpoles, we included some water from the sample site in the bags. Once the sample size for that day was reached or it was noon, the amphibians were weighed in the Whirl-Pak bags. Wearing sterile nitrile gloves, we then removed each amphibian from its bag and weighed the Whirl-Pak bag by itself, with amphibian mass as the difference. The amphibian was then rinsed with 50 ml of autoclaved distilled water to remove dirt and environmental bacteria. We then swabbed amphibians using a single sterile rayon swab (MW113, Medical Wire Equipment, Corsham, England). Swabbing of adults consisted of five strokes in one direction on each of the hind feet and thighs, and 20 strokes on the ventral side (total strokes = 40), while for tadpoles, we swabbed around their mouth 25 times, an area commonly infected by Bd in this lifestage (*Marantelli et al., 2004*). We then placed the swab in a sterile 1.5 ml microcentrifuge tube and placed it on ice while in the field. Lastly, we measured the snout-vent length of the adult frogs, and the tail length and total length of tadpoles, before they were released back in the site. Upon returning from the field (<6 h after sampling), we stored the swabs in a −20 °C freezer. At the end of June, all samples were transferred to Virginia Tech, Blacksburg, VA, and stored in a −80 °C freezer.

### Environmental data collection

We took temperature and pH measurements (Oakton Waterproof pHTestr 30) at locations where amphibians were sampled each day. Water temperatures (overall mean = 19.7, sd = 3.51) were similar across the different sampling locations (Fig. S1A; Fig. S2A). The pH also was similar across all sampling sites (all with mean ~7), except the vernal pool subsite at site 1, which had a lower mean pH of 4.75 (Fig. S1B; Fig. S2B).

### Sample and data processing

All amphibian samples, across all three sites, were used to determine the prevalence of Bd in MRGP. However, bacterial community data was only obtained from 2017 samples

across all five subsites at site 1, and consisted of 66 swabs from three amphibian species; *L. sylvaticus*, *L. clamitans*, and *L. palustris* (Table 1).

### Swab DNA extraction

The swabs were transported to Virginia Tech in Blacksburg, VA, where they were all processed by a single individual (ZG). DNA was extracted from the swabs using a Qiagen DNeasy Blood and Tissue Kit (Qiagen, Hilden, Germany) with the lysozyme pre-treatment for gram-positive bacteria. The lysozyme pre-treatment for each sample consisted of adding 0.18 ml of lysis buffer and 3.7 g of lysozyme, vortexing, and incubating for one hour at 37 °C. We then stored the extracted DNA in 100 µl of molecular grade water in a −20 °C freezer, and used it as template DNA for Bd PCR and assessing skin bacterial communities using 16S rRNA gene amplicon sequencing.

### Bd detection

Following DNA extraction, we screened all the amphibian swabs collected (2017 and 2018, N = 408) for Bd using PCR (*Annis et al., 2004*). Each 25 µl PCR reaction, one per sample, contained: 0.5 µl of dNTPs, 0.2 µl of Taq DNA Polymerase, 3.9 µl of Taq Buffer with MgCl$_2$, 2.5 µl of ITS 1–3 primer, 2.5 µl of 5.8S primer, 2 µl of extracted sample DNA, and 13.4 µl of water. The thermocycler conditions were: 93 °C for 10 min to start, then 30 cycles of 93 °C for 45 s, 60 °C for 45 s, and 72 °C for 1 min, and a final 10 min at 72°C. Every thermocycler run had a positive (extracted DNA from a Bd JEL 404 stock) and negative (molecular grade water) control. Each PCR product was run on a 1% TAE agarose gel. The gels were inspected to ensure the amplification in the positive sample and no amplification in the negative sample. We recorded a sample as Bd positive if a band was seen on the gel.

### Skin bacterial communities

To assess the skin microbiome, we completed 16S rRNA gene amplicon sequencing of the 66 samples collected in 2017 at site 1 (17 *L. sylvaticus*, 25 *L. palustris*, 12 *L. clamitans*, and 12 *Lithobates* tadpoles). We amplified the V4 region of the 16S rRNA gene using the 515F and 806R primers. The 806R primer contained a 12bp error-correcting Golay code to mark individual samples. Each 25 µl reaction contained 0.5 µl of both the 515F and 806R primers (at 10 µM concentration), along with 12 µl of ultra-clean PCR grade water, 10 µl of 5Prime hot master mix, and 2 µl of the template DNA. Each 25 µL PCR reaction was run in triplicate, along with a negative control reaction that did not include any template DNA. The thermocycler conditions were: 94 °C for 3 min to start, then 35 cycles of 94 °C for 45 s, followed by 50 °C for 1 min, and 72 °C for 1.5 min, and a final 10 min at 72 °C. At the end of the PCR run, we combined each sample's triplicate PCR products into one tube. We visualized the combined PCR product on a 1% TAE agarose gel, where the negative samples were checked for contamination and the sample wells were checked for amplification. The PCR products were quantified using a Qubit 2.0 fluorometer (Invitrogen, Carlsbad, CA, USA) with a dsDNA High Sensitivity assay kit. We combined 200 ng of DNA from each sample (N = 66) into a final pool. We then purified the pooled sample using the QIAquick PCR purification kit (Qiagen, Hilden, Germany) and sent it to

the Genomics Center at the Dana Farber Cancer Institute of Harvard University for 250 bp single-end sequencing on an Illumina Mi-Seq instrument.

### Sequence processing

We processed the 250 bp forward (single-end) reads using the QIIME2 pipeline (*Bolyen et al., 2018*). We imported the raw reads and demultiplexed them. The reads were of consistently high quality across the full-length, so we did not need to trim them. We denoised the data using DADA2 (*Callahan et al., 2016*), which included filtering out phiX and chimeric reads, as well as correcting amplicon errors. We used the recommended 'big data' parameters for DADA2, which included truncating reads with a quality score cut-off of 11. In addition, we only used 10,000 reads to build the error distribution, which we have found is adequate for our high-quality data and significantly decreases the run time. The resulting amplicon sequence variants (ASV) table contained 45,080 ASVs. We then filtered out any ASVs that were present at less than 0.01% of the total read count, to work with ASVs that are likely more impactful in the system. This left 1,103 ASVs in the table. Taxonomy was assigned to these ASVs using the SILVA v132 database (*Quast et al., 2012*) and scikit-learn classifier (*Pedregosa et al., 2011*). We then filtered out any ASVs that were assigned as chloroplasts, mitochondria, or were unassigned; this cut the number of ASVs to 1,079. After visualizing the alpha rarefaction curve, the dataset was rarefied at 20,000 reads, which resulted in the loss of four samples with lower read counts (three *L. sylvaticus* and one *L. palustris*). The final table contained 62 samples (Table 1) and 1,079 ASVs.

## Statistical methods

We calculated two metrics in QIIME2 (*Bolyen et al., 2018*) to estimate within sample diversity (alpha diversity): Shannon diversity and ASV richness. Shannon diversity, a metric that accounts for both richness and evenness of ASVs, and ASV richness, the number of different ASVs in the sample, are common alpha diversity metrics used in bacterial communities studies (*Buttimer, Hernández-Gómez & Rosenblum, 2021*; *Jani et al., 2021*). All data were exported from QIIME2 and analyzed in *R* v.4.0.0 (*R Core Team, 2020*) unless otherwise specified. All statistical tests, unless mentioned otherwise, used data from both adults broken up by species, and Lithobates tadpoles.

### Differences in bacterial communities between amphibian species and life stage and sampling subsites

To assess differences in the community structure of ASVs across amphibian species and life stage, the skin bacterial community data was visualized with non-metric multidimensional scaling (NMDS) plots using a Bray-Curtis dissimilarity index (Fig. 2A) and Jaccard dissimilarity index (Fig. 2B). The Bray-Curtis dissimilarity index uses relative abundance data and the Jaccard dissimilarity index uses presence and absence data. Skin bacterial communities were also visualized using a Bray-Curtis dissimilarity index and compared across subsites, were the amphibians were collected from (Fig. 3). We used two permutational multivariate analysis of variance (PERMANOVA; *vegan*: *adonis*; *Oksanen et al., 2013*) to test if the multivariate means (centroids) and variances (dispersion) of the

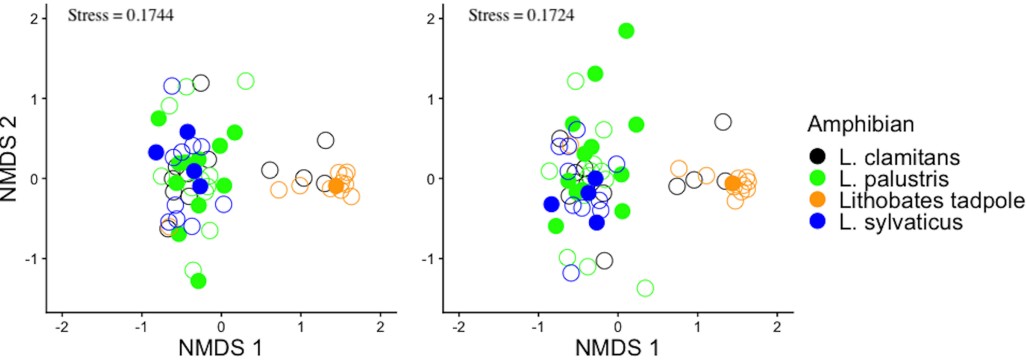

**Figure 2 Bray-Curtis and jaccard dissimilarity matrix NMDS plots of amphibian skin bacterial communities.** The bacterial data was plotted twice with non-metric multidimensional scaling (NMDS) plots. The plots show all amphibian bacterial samples plotted with an (A) Bray-Curtis dissimilarity matrix and (B) Jaccard dissimilarity matrix. Amphibian samples were colored by adult Lithobates species and Lithobates tadpoles. Also plotted is the amphibian's infection status shown by filled circles (infected) and empty circles (uninfected).

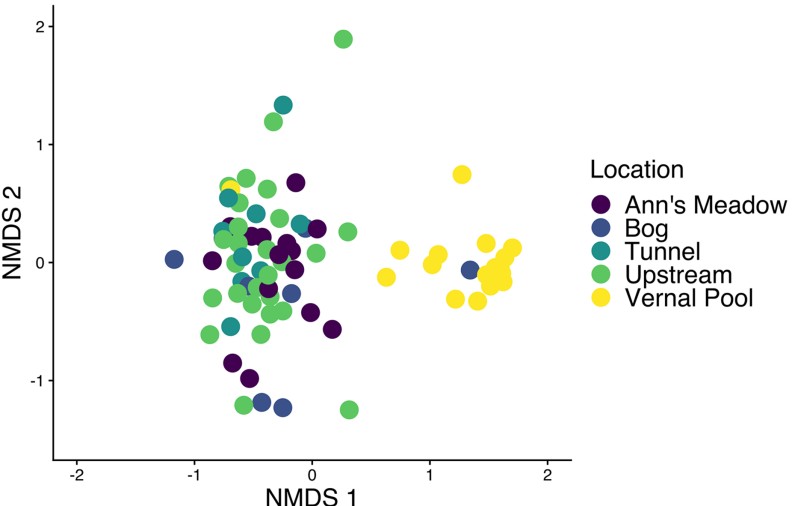

**Figure 3 Amphibian skin bacterial samples collected from different locations.** All amphibian bacterial samples are shown plotted with an NMDS using a Bray-Curtis distance matrix. All samples are colored based on where the sample was collected from at site 1. The stress value is 0.1744.

skin bacterial samples, grouped by (1) amphibian species and life stage and (2) subsites, were different. The subsite PERMANOVA included all subsites besides the vernal pool, where the majority of samples were tadpoles, and therefore, development was a confounding variable. Using the *pairwise.adonis* function, we then performed a *post-hoc* test on the two PERMANOVAs to determine which species or life stage and subsites were different from each other (*Martinez Arbizu, 2020*). To check the homogenous dispersion among (1) host species and life stage and (2) subsites, we used the *betadisper* function which calculates the samples distance from the centroid and uses a permutation approach to tests for statistical differences (*vegan: permutest*; *Oksanen et al., 2013*). Again, the vernal

pool subsite was not included in the dispersion analysis due to the majority of samples being from tadpoles. Each permutation test was run twice, once for each dissimilarity matrix, with 999 permutations for each test. Lastly, we ran two linear discriminate effects size analyses (LEfSe), using the *run_lefse* function in the *microbiomeMarker* package (*Cao et al., 2022*), to compared bacteria phylum abundances across (1) adults species and tadpoles and (2) sampling subsites.

To determine how bacterial diversity measurements (Shannon diversity and ASV richness) differed across amphibian species and between life stages and sampling sites the five subsites in site (1), we used a series of generalized linear models (GLMs). For each alpha diversity measurement, we fit two models, one for subsites at site 1 and one for amphibian species and life stage, for a total of 4 GLMs. Shannon diversity GLM models were fit assuming a Gamma distribution with an inverse link function as these metrics are continuous and strictly positive. The ASV richness metrics were fit with a negative binomial GLM with a log link function since this metric is discrete data and overdispersed. After fitting the models, we used the *emmeans* function from the *emmeans* package (*Lenth et al., 2018*) to make contrasts, and used the Tukey method to adjust for multiple testing between adult amphibian species and lifestage, and sampling sites for each of the metrics independently.

### Differences in species and site infection prevalence

We ran a logistic regression to compare adult species and tadpole's infection prevalence. Using the *emmeans* function from the *emmeans* package (*Lenth et al., 2018*), we made contrasts to compare species and tadpoles. Additionally, we ran a mixed effect logistic regression, with species as a random effect and site as a fixed effect, using the *glmTMB* function from the *glmTMB* (*Brooks et al., 2017*). Lastly, we again ran contrasts with the *eemeans* package (*Lenth et al., 2018*) to compare sites to each other.

### Differences between bacterial communities in infected and non-infected *Lithobates palustris*

To compare bacterial communities between infected and non-infected amphibians, we only used *L. palustris* since this was the only species with enough samples of infected individuals (2017: N = 12 infected, 13 uninfected) for a meaningful statistical comparison. Differences in bacterial communities between infected and non-infected *L. palustris* were compared using logistic GLMs with alpha diversity measures set as predictor variables. Each alpha diversity measure addresses a different aspect of diversity (but are correlated) and therefore only one was used in a model at a time. We also examined the bacterial community structure between infected and non-infected individuals using a PERMANOVA and permutation test, as described above.

## Results

We identified a total of 1,079 unique ASVs and out of these there were 946 ASVs present on *L. clamitans*, 893 ASVs on *L. palustris*, 769 ASVs on *L. sylvaticus* and 811 ASVs on *Lithobates* tadpoles. The phylum Proteobacteria dominated most of the amphibian bacterial samples, with 596 ASVs and ~60% of the total relative abundance in all of the

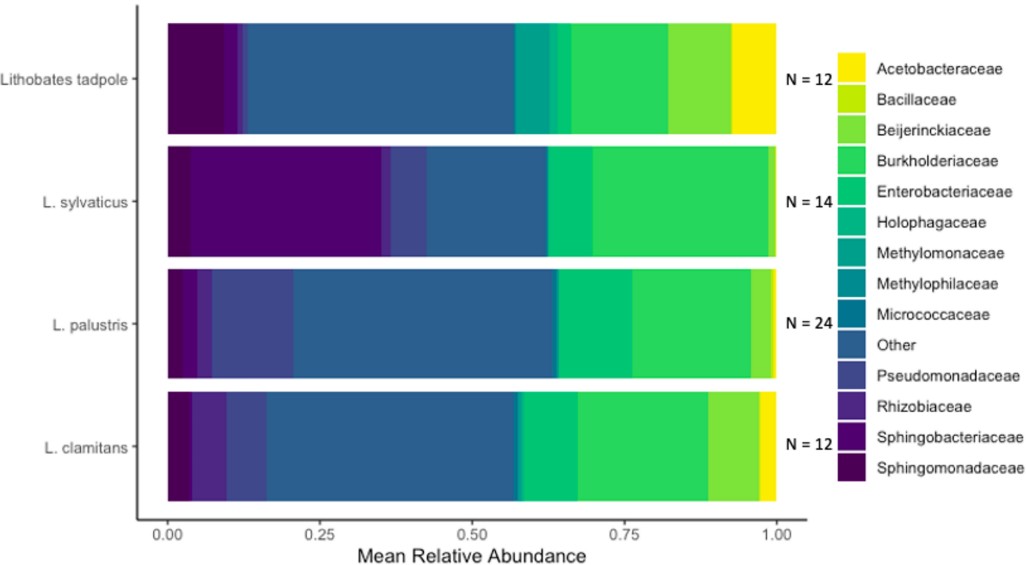

**Figure 4 Bacterial families found in amphibian skin samples.** The mean relative abundance of bacterial families found on amphibian bacterial swabs. Families that had less than 5% mean relative abundance were grouped into the other category. The sample size used to create the mean relative abundance bars for each group is shown at the end of each bar.

samples. Bacteroidetes was a abundant phyla in *L. sylvaticus* samples with 126 ASVs and >12% relative abundance in these groups (LEfSe; coefficient = 5.25, *p*-value < 0.001), however, this phylum was lower in the rest of the amphibian samples. Additionally, the phyla Tenericutes was more abundant in *L. sylvaticus* samples (LEfSe; coefficient = 3.96, *p*-value < 0.001). Abundant phylum in tadpoles samples included Acidobacteria (LEfSe; coefficient = 4.29, *p*-value < 0.001), Verrucomicobia (LEfSe; coefficient = 4.06, *p*-value = 0.001), Spirochaetes (LEfSe; coefficient = 3.42, *p*-value < 0.001), Cyanobacteria (LEfSe; coefficient = 3.03, *p*-value < 0.001), and Fibrobacteres (LEfSe; coefficient = 2.81, *p*-value < 0.001). The phyum Gemmatimonadetes was abundant in *L. clamitans* (LEfSe; coefficient = 2.84, *p*-value = 0.044) and Planctomycetes was abundant in *L. palustris* (LEfSe; coefficient = 3.36, *p*-value = 0.005). Common bacterial families included Burkholderiaceae, Enterobacteriaceae, and Beijerinckiaceae with a mean relative abundance of over 10% in multiple amphibian groups (Fig. 4).

Abundant phylum at the vernal pool subsites were similar to those of tadpole samples (Acidobacteria, Verrucomicrobia, Spirochaetes, Cyanobacteria, and Fibrobacteres; LEfSe; all *p*-values < 0.05). In addition to those phylum, WPS-2 was also abundant in vernal pool subsite samples (LEfSe; coefficient = 2.98, *p*-value < 0.001). Actinobacteria was abundant in bog subsite samples, and Planctomycetes were abundant in tunnel subsites samples. Lastly, Bacteroidetes (LEfSe; coefficient = 4.96, *p*-value =0.045), Tenericutes (LEfSe; coefficient = 3.78, *p*-value = 0.006), and Gemmatimonadetes (LEfSe; coefficient = 2.66, *p*-value = 0.016) were all abundant in upstream subsite samples, and Ann's Meadow had no abundant distinguishing phylum.

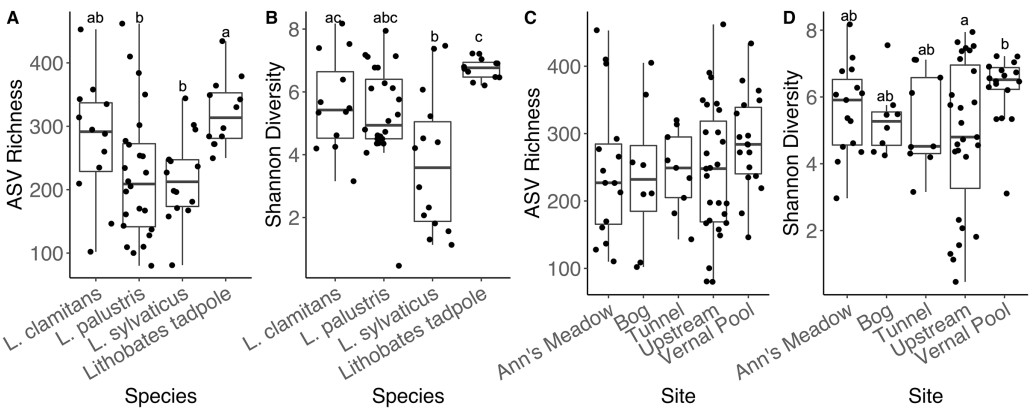

**Figure 5 Alpha diversity plots for sampling locations and amphibian species.** Alpha diversity measurements, ASV richness and Shannon diversity, across species and developmental stage (A and B), and across subsites at site 1 (C and D).

### Differences in bacterial communities between amphibian species and life-stages

Adult amphibians had similar ASV richness (all *emmeans* contrasts *p*-values > 0.05). We found *L. clamitans* had a significantly higher Shannon diversity (Fig. 5B) than *L. sylvaticus* (coefficient = 0.0894, SE = 0.0310, Z.ratio = −2.882, *p*-value = 0.018), but both species had similar Shannon diversity compared to *L. palustris* (all contrasts *p*-values > 0.05). Lastly, when we compared bacterial communities of adults to tadpoles, we found that tadpoles had a higher ASV richness than the adults from two species (tadpoles-*L. palustris*; coefficient = 0.3671, SE = 0.128, Z.ratio = 2.873, *p*-value = 0.0212; tadpoles-*L. sylvaticus*; coefficient = 0.3889, SE = 0.142, Z.ratio = 2.733, *p*-value = 0.0319; other *p*-value > 0.05), and a higher Shannon diversity than the adults of one of these species (tadpoles-*L. sylvaticus*; coefficient = 0.1187, SE = 0.0294, Z.ratio = 4.039, *p*-value < 0.001; all other *p*-value > 0.05).

Bacterial community structure was different between amphibian species and life stages (Fig. 2; PERMANOVA Bray-Curtis: $F_{(3,58)}$ = 4.8837, $R^2$ = 0.202, *p*-value = 0.001; Jaccard: $F_{(3,58)}$ = 4.488, $R^2$ = 0.188, *p*-value = 0.001). Specifically, we found that *L. sylvaticus* bacterial community structure was different than both *L. palustris* (*p*-value = 0.006) and *L. clamitans* (*p*-value = 0.006). But, *L. palustris* and *L. clamitans* had similar bacterial community structures (*p*-value > 0.05). Additionally, we found that tadpole bacterial community structures were different than all adult amphibians species (all *p*-values < 0.01). Further, we found that dispersion was significantly different among amphibians species and life stage (*permutest* Bray-Curtis: $F_{(3,58)}$ = 7.305, *p*-value = 0.001; Jaccard: $F_{(3,58)}$ = 7.305, *p*-value = 0.001). We found that *L. palustris* had a greater dispersion, a greater variation in bacterial communities among samples, than *L. sylvaticus*, and *L. clamitans*, and *L. palustris* also showed a greater dispersions than tadpoles. This potentially indicates that *L. palustris* bacterial community structural differences might be due to differences in dispersion rather than centroids, multivariate means.

### Differences in species and site infection prevalence

Infection prevalence was greatest in *L. palustris* at site 1, with 48% infected in 2017 and 59% infected in 2018 (Table 1). From the logistic regression, we found that infection prevalence was greater in *L. palustris* compared to *L. sylvaticus* (coefficient = −2.090, *p*-value < 0.001), *L. clamitans* (coefficient = −3.411, *p*-value < 0.001), and tadpoles (coefficient = −1.629, *p*-value < 0.001). However, *L. palustris* infection prevalence was not significantly greater than *L. catesbeianus* (coefficient = −2.170, *p*-value = 0.2454), which was only sampled at one site and year. Lastly, when we used species as a random effect in a logistic regression with site as a fixed effect we found that site 2 had a higher infection prevalence than both site 1 (coefficient = 1.64, *p*-value = 0.0014) and site 3 (coefficient = 3.23, *p*-value < 0.001). While, site 1 had a higher infection prevalence than site 3 (coefficient = 1.60, *p*-value = 0.0054).

### Differences between bacterial communities in infected and non-infected *Lithobates palustris*

A comparison between the infected and non-infected *L. palustris* bacterial communities suggested no differences in any of the alpha diversity measurements (Shannon diversity: glm, b = 0.2032, SE = 0.2854, Z = 0.712, *p*-value = 0.467; ASV richness: glm, b = −0.0005, SE = 0.0041, Z = −0.122, *p*-value = 0.903). We also did not find any differences in bacterial community structure (Figs. 2A and 2B) between infected and non-infected *L. palustris* (PERMANOVA Bray-Curtis: $F_{(1,22)}$ = 0.717, $R^2$ = 0.032, *p*-value = 0.871; PERMANOVA Jaccard: $F_{(1,22)}$ = 0.832, $R^2$ = 0.0362, *p*-value = 0.873).

### Differences between amphibian bacterial communities across site 1

We found that adult amphibian skin bacterial community structured was different across sampling sties (Fig. 3; PERMANOVA Bray-Curtis: $F_{(3,43)}$ = 2.2579, $R^2$ = 0.1340, *p*-value = 0.001; Jaccard: $F_{(3,43)}$ = 1.305, $R^2$ = 0.092, *p*-value = 0.002). Adult amphibians found at the bog site had a difference skin bacteria structure than adult amphibians from other subsites (all *p*-values < 0.05). Additionally, adults sampled from the tunnel subsite had different skin bacterial community structures than adults sampled form the Ann's meadow subsite (*p*-value = 0.012). Lastly, we found no significant difference in dispersions between subsites were adult skin bacterial communities were sampled (Bray-Curtis: $F_{(3,43)}$ = 2.3994, *p*-value = 0.089; Jaccard: $F_{(3,43)}$ = 1.018, *p*-value = 0.387). The vernal pool subsite was not included in the subsite beta diversity analysis due to the majority of the samples being from tadpoles, and thus, development and site are confounded. However, in the subsite NMDS, *Lithobates* tadpoles and *L. clamitans*, collected in or around the vernal pool, clustered together, and separately from the other samples (Fig. 3). Lastly, we found that amphibians in site 1 at different subsites had similar skin bacterial ASV richness and Shannon diversity (all *GLM p*-values > 0.05; Figs. 5C and 5D).

## DISCUSSION

In this study, we found amphibian skin bacterial communities and Bd infection prevalence varied across Ranid frog species at the MRGP. We expected that infection prevalence to be low throughout the preserve, and for any infections to be uniform across all sampled

species. However, we found that *L. palustris* had a higher infection prevalence than other species at one of the sites sampled. Additionally, we found that when we sampled *L. palustris* at this site the next year, infection prevalence remained higher compared to other sites and other amphibian species. Our infection results might be biased towards individuals with higher infection intensity due to out use of PCR and gel bands to identify infected individuals. The use of qPCR might have identified individuals with lower infection intensity. However, our results were similar to a study that surveyed amphibians in the southeastern United States, where *L. palustris* had a higher Bd infection prevalence than other amphibians surveyed (*Rothermel et al., 2008*). A survey in Connecticut, close to our sampling sites, found that Bd infection prevalence in *L. palustris* was similar to *L. clamitans*, although this study only had 18 *L. palustris* samples compared to 266 *L. clamitans* samples (*Richards-Hrdlicka, Richardson & Mohabir, 2013*). Our results, along with results from the literature, show how variable Bd infection can be across different sampling sites, even those that are in close proximity. This result indicates the importance of sampling over a wide set of sites in an area to determine Bd dynamics and prevalence.

We also predicted that infected individuals would have different bacterial communities than non-infected individuals, but we did not find significant differences in bacterial communities between infected and non-infected *L. palustris*. Experimental trials have shown that skin microbial communities change with Bd infection and, in field studies, skin bacterial diversity can differ between Bd positive and negative sites (*Jani & Briggs, 2014*; *Rebollar et al., 2016*). With field studies, we do not know about previous infection history and whether some of our uninfected individuals may have already been infected and cleared the infection. Previous history with Bd might have already shifted the amphibian skin bacterial communities, and therefore, we are not finding a significant differences. Additionally, we are focused on three species of amphibians that are not particularly susceptible to Bd and we might expect to see more drastic bacterial community shifts with more susceptible species. Other studies have not found differences in skin bacterial communities based on Bd infection status and have made similar suggestions (*Kruger, 2020*; *Belden et al., 2015*). However, we only used one amphibian species to compare bacterial communities due to low infection prevalence across all other species, and Bd might impact host skin microbial communities differently in other species. Additionally, we did not test for infection intensity, and the severity of Bd infection can affect bacterial communities (*Jani & Briggs, 2018*). Therefore, if infected individuals in our study had low infection intensity, there might not be an impact on bacterial communities, or the impact on bacterial might be too small to identify with our sample size.

We did find evidence that *L. sylvaticus* had different bacterial communities than *L. clamitans* and *L. palustris*. Previous studies have also found species that coexist at the same site had different skin bacterial communities, suggesting that there might be some host-specific factors that influence what bacteria are becoming members of the skin bacterial community, instead of amphibians passively collecting bacteria from their environment (*Buttimer, Hernández-Gómez & Rosenblum, 2021*; *Abarca et al., 2018*; *Belden et al., 2015*; *Kueneman et al., 2014*; *McKenzie et al., 2012*; *Walke et al., 2014*). However, other studies have found that the association between host phylogeny and differences in

skin bacterial communities are weaker at lower taxonomic groupings (*i.e.*, genus and species), suggesting that habitat, along with host life history, may be a better predictor of bacterial community differences (*Ellison et al., 2019*; *Bird et al., 2018*; *Bletz et al., 2017*). Interestingly, while both *L. clamitans* and *L. palustris* were primarily caught in water bodies (*i.e.*, bogs, streams, and vernal pools), *L. sylvaticus* was primary caught on land. This potentially indicates a link between habitat usage and amphibian skin bacterial communities, however the effect of species and habitat cannot be separated here. Additionally, when comparing bacterial communities between adult amphibian sampling locations, we find that amphibian skin bacterial communities from the bog sampling site were different from other sampling locations. Interestingly, this area was the most open, in terms of forest cover, and had the most distinct difference in vegetation compared to other sampling locations.

We also found evidence of differences between tadpoles and adult frogs. There were differences in both alpha (*i.e.*, ASV richness) and beta diversity metrics between adults and tadpoles, though this pattern was not as clear when using the Shannon index. Overall, this result suggests that bacterial communities may change due to metamorphosis in these species, as has been found previously (*Kueneman et al., 2014*; *Prest et al., 2018*). Even though we found these differences between tadpoles and adult amphibians, we swabbed different parts of their body: mouthparts for tadpoles and whole bodies for adult amphibians. Amphibian skin microbial communities can differ across body regions (*Bataille et al., 2016*), and this might contribute to some of the differences between adult and tadpole bacterial communities. Additionally, all collected tadpoles were from the vernal pool subsite, which had a lower pH than the surrounding wetlands. This environmental difference could also have led to some of the differences in the bacterial communities we saw between tadpoles and adults. Environmental factors, such as temperature, salinity, elevation, precipitation, and pH can shape bacterial communities (*Estrada et al., 2019*; *Varela et al., 2018*; *Albecker, Belden & McCoy, 2019*; *Hughey et al., 2017*; *Longo & Zamudio, 2017*). Thus, we cannot rule out alternative explanations of the observed differences in bacterial communities.

## CONCLUSION

Our results highlight potential developmental and species differences in amphibian skin bacterial communities and that Bd infection varies among species and sites. Collecting more data on a variety of amphibian hosts in different environments will help tease out the relative contributions of environmental, host, and pathogen characteristics to the amphibian skin microbiome. This will allow us to determine when certain factors might influence host microbial communities and to what extent. Host microbiomes can be critically important for host disease susceptibility and understanding what characteristics shape microbiomes could provide insights into conservation efforts in a variety of wildlife disease systems.

### Funding

This work was supported by the Mianus River Gorge Preserve Research Assistantship Program. Additionally, Zachary Gajewski was partially supported by NIH EEID R01A122284 and Leah R. Jonhson by NSF DMS/DEB 1750113. After receiving funding from the Mianus River Gorge Preserve (MRGP), we worked with Christopher Nagy (a coauthor and Director at MRGP) to design and write up this study and manuscript. The funders had no role in data collection and analysis, and decision to publish.

### Grant Disclosures

The following grant information was disclosed by the authors:
 Mianus River Gorge Preserve Research Assistantship Program.
NIH EEID R01A122284.
NSF DMS/DEB 1750113.

### Competing Interests

Christopher Nagy is the Director of Research & Education at the Mianus River Gorge Preserve.

### Author Contributions

- Zachary Gajewski conceived and designed the experiments, performed the experiments, analyzed the data, prepared figures and/or tables, authored or reviewed drafts of the article, and approved the final draft.
- Leah R. Johnson conceived and designed the experiments, analyzed the data, prepared figures and/or tables, authored or reviewed drafts of the article, and approved the final draft.
- Daniel Medina conceived and designed the experiments, performed the experiments, prepared figures and/or tables, authored or reviewed drafts of the article, and approved the final draft.
- William W. Crainer conceived and designed the experiments, performed the experiments, prepared figures and/or tables, authored or reviewed drafts of the article, and approved the final draft.
- Christopher M. Nagy conceived and designed the experiments, performed the experiments, prepared figures and/or tables, authored or reviewed drafts of the article, and approved the final draft.
- Lisa K. Belden conceived and designed the experiments, prepared figures and/or tables, authored or reviewed drafts of the article, and approved the final draft.

### Animal Ethics

The following information was supplied relating to ethical approvals (*i.e.*, approving body and any reference numbers):
    Animal care protocols were approved by Virginia Tech's Institutional Animal Care and Use Committee (16-193-STAT).
## Field Study Permissions

The following information was supplied relating to field study approvals (*i.e.*, approving body and any reference numbers):

Amphibian collection was approved by Mianus Rover Gorge Preserve and the state of New York (Permit #2213).

## DNA Deposition

The following information was supplied regarding the deposition of DNA sequences:

The pooled Illumina sequence reads are available at GenBank: PRJNA936869.

The sequence data is also available in the Supplemental File.

## Data Availability

The analysis and figure code, ASV table, alpha diversity measurements, and amphibian infection data are available at Zenodo: gzach93. (2023). gzach93/ MianusRiverGorge_RanidFrogs: Mianus River Gorge Ranid Frogs (7May23). Zenodo. https://doi.org/10.5281/zenodo.7905229.

## Supplemental Information

Supplemental information for this article can be found online at http://dx.doi.org/10.7717/ peerj.15556#supplemental-information.

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
