# Peer review of "Skin bacterial community differences among three species of co-occurring Ranid frogs"

_PeerJ, doi:10.7717/peerj.15556_

## Round 0.1 · original submission · Major Revisions

Based on the comments provided by the 3 reviewers, there is room for improvement of your description of methods, as well as in the way some results are analyzed and/or presented. I encourage you to take into account all the suggestions and provide an updated version of your work upon responding to these recommendations.

Reviewer 1 ·

Basic reporting

Skin microbial communities play critical roles in maintaining host health and serve as the first line of defense against pathogens. While many studies have examined how skin microbial communities protect the host from fungal infections, there remain questions on how host species and environmental factors shape microbial communities. In this study, Gajewski et al., addressed this knowledge gap by sampling skin microbiomes from three species of adult frogs and tadpoles from three different locations. They identified inter-species, inter-individual, and inter-habitat variations between amphibian skin microbial communities and Batrachochytrium dendrobatidisinfection infection rates, providing insights into characteristics of amphibian skin microbial communities. The manuscript is well-structured and well-written. The result is well-demonstrated with appropriate description, thus I would support the publication of this work if the authors could address some minor points below:

Introduction:
Line 61: “declines globally” -> “declining globally”.
Line 65-68: please provide more than one example of the relations of the current researches.

Experimental design

Methods:

It’s exciting to see there are microbial differences across different species, developmental stages, and sample locations. However, it would be better if the authors can perform correlational analysis on different taxonomic levels, such as species, genus, or family levels, to point out if any taxon’s abundances are highly correlated with their host species, developmental stages or sample locations. One example of such analyses I highly recommend would be Linear discriminant analysis Effect Size (LEfSe) analysis. I would expect to see some species are highly enriched in adults vs tadpoles or in vernal pools.

Validity of the findings

Results:

Please include panel labeling “A” and “B” for figure 3.


Data accessibility

Please deposit your raw data of 16S rRNA sequencing to a public data repository (such as NCBI SRA), thus other researchers could potentially apply it for other analysis.

Additional comments

no comment

Reviewer 2 ·

Basic reporting

In this study the authors seek to determine differences in microbiome composition and Bd infection between three closely related species of frog. Additionally, authors attempt to compare adult and tadpole microbiomes. The authors have done a good job with clearly a tough system to sample. Overall, I find this study to be valuable and suitable for publication after some revision.

Experimental design

There are many issues with the experimental design that, while can’t be addressed, should be discussed in more detail for transparency. Particularly, the issue with using gel bands to infer Bd prevalence needs to be laid out in the methods and discussion. Sample sizes are small and sporadic so a clearer explanation of the sampling design, including an improved figure, would be valuable. Finally, I think more discussion on habitat differences would provide the something extra that this paper is currently lacking.

Validity of the findings

I feel that using gel bands to infer Bd prevalence is a major issue with this paper, as there is a high likelihood of false negatives. This is likely skewing the prevalence metric toward high load detection only.

Additional comments

Below I provide line-by-line comments on the different manuscript sections.

Introduction:
Lines 48-56: In light of your results, I think it makes more sense to frame the introduction around environmental influences on host microbiomes instead of microbiomes influencing host health (that should come after). You could start this paragraph with something about host microbiomes being dynamic and shaped by environmental and host factors.
Line 58: “System” is repeated, reword.
Lines 68-71: This sentence could use rewording, maybe add “…has inspired a number of studies…”
Line 73: Try to be consistent with your use of “Bd”, I think the use of “chytrid” sounds a little casual.
Line 73-74: I don’t think this is completely true, we don’t understand all the factors, or maybe be even most of the factors, but we definitely understand some factors. Lots of recent literature showing the effects of habitat disturbance and host identity on microbiome composition here. Additionally, I don’t think your study is set up to answer the actual mechanisms driving microbiome composition, so it is important to avoid pointing to a gap in the literature that you aren’t able to address, at least here in the intro.
Lines 74-76: I think this sentence sounds a little weak after some of the previous statements, consider rewording the first part.
Lines 78-80: This sentence is pretty long and repetitive, I think there is a lot of extra stuff you could toss out that you can detail in your methods. For example, I think you can remove “un-manipulated” and the first instance of “Bd infection” and “skin microbiome” since those are repeated.
Line 84: I think adding bullfrogs as a fourth species makes things more confusing here, so either add it with the other 3 species above or just mention its inclusion in the methods.
Line 90: There is nothing about habitat differences here, which seem to be a focus of this study. Consider reworking your predictions to include habitat.

Methods:
Lines 95-113: This sampling design is a little convoluted, which is understandable in a field study, but I think a comprehensive figure showing a map of sample sites, what year each site was sampled, what species were sampled, and what samples were collected would be useful for readers.
Line 127: Replace “in tadpoles” with “in this lifestage”.
Line 134: It would be a good idea to indicate the range of detected temps here, since that is important for Bd infection, particularly since you sampled in June.
Line 136: A pH of 4.75 is extremely acidic, did you double check these values?
Lines 139-142: I think this has already been stated, although I do like how clear it is stated here. I think with a nice figure detailing sample design you can delete these sentences.
Line 147: You should give more details on the lysozyme pretreatment since it isn’t a standard part of the DNeasy kit.
Lines 153-162: Using PCR gel bands to detect Bd infection seems highly susceptible to false negatives. In general, particularly in the summer, Bd loads are relatively low which would be hard to see on a gel. Additionally many factors can influence whether or not gels show bands, even when the positive control is working properly. With microbiome samples we often see no band but get plenty of reads back after sequencing. I think there needs to be a clear statement made addressing the potential errors in Bd prevalence due to these methods.
Lines 168-169: Be consistent on whether you are adding a space between numbers and units or not (“12bp” and “0.5 uL”).
Line 193: How did you decide on the 0.01% threshold? I know Bokulich recommends a 0.005% which is arbitrary, but probably worth explaining how you decided on this number.
Line 198: Very impressive!
Lines 216-219: Could you also test for differences between sites here?
Line 230: I think, for the site model, it might be interesting to run all data adult together and include host identity as a random effect.
Line 233: Wouldn’t a Poisson distribution be more appropriate?

Results:
Line 258: “phyla” should be singular here, “phylum”.
Lines 267-274: You should report more than just the p-value, consider adding supplementary tables.
Lines 288-293: I am not convinced that this analysis is adding all that much to the paper. Maybe if there was a functional role of these bacteria known, but as is this information doesn’t seem very relevant. Maybe use the indicator species analysis between infected and uninfected L. palustris, then check for anti-Bd function in the Woodhams database?
Lines 311-313: This is very interesting! Especially since in general tadpole microbiomes clustered separately from adult microbiomes.

Discussion:
Line 316: I think you could start a little punchier here “In this study we found species-level differences in Bd infection and microbiome composition of Ranid frogs.” Or something like that to really set the tone of the discussion.
Line 319: I would argue that L. palustris only had higher “high load” prevalence, since you are likely not detecting low-load infections. It is important to discuss the limitations of your metric of Bd infection in this or the next paragraph of the discussion.
Line 343: Again, you need to discuss that you are likely missing low-load infection which affects your metric of prevalence. You should consider reframing discussion of prevalence as “high load prevalence”.
Lines 359: You should discuss the differences between your subsites, what habitat differences could be driving differences in prevalence and potentially microbiome composition (if you run those permanova).
Line 372: Maybe see if any of the detect indicator ASVs on tadpoles are associated with acidic environments?
Line 375: And habitat disturbance, I think some discussion on how pristine your sites were, an indication of forest cover, and how that is expected to affect microbiome similarity and Bd prevalence is needed.

Figures:
Figure 1: consider revising this map as indicated in my comments in the methods.
Figure 3: This is too much information on one plot, separate the host comparison, life stage comparison, and infection status comparison onto different plots. Also consider changing to softer colors.
Figure 4: I think displaying all the points over the boxplots would be valuable in this case since sample sizes were variable between sites.
Figure 5: I love this figure, and I think the difference between the vernal pool and the other sites should be a focus of your results and discussion. Try to reveal what taxa are driving these differences (LEfSe/ANCOM, indicator species analysis, Venn Diagram).

Reviewer 3 ·

Basic reporting

This manuscript investigates the skin bacterial communities of three amphibian species. The aim of this manuscript is to determine links between skin bacterial communities and Batrachochytrium dendrobatidis, chytrid, in Lithobates palustris. Additionally, this study seeks to tease apart the varying impacts of host and environmental variation on skin microbiome variation in four Lithobates species. The authors find no evidence for chytrid infection status determining skin microbiome variation. However, skin microbiomes are found to be influenced by host species, host life stage and habitat type. Because different species were caught in different habitat types the results might indicate differences in habitat use by adult amphibians to be the main driver of skin bacterial differentiation.

The manuscript is clear and well written - I really enjoyed reading it! The structure and research ideas are well defined and clear from the Introduction to the Discussion. The authors also have a nice and extensive dataset of Lithobates host and environmental data, including an adequate skin microbiome data set for Lithobates palustris. This data must have taken a lot of effort to compile across sites and over years.

Experimental design

The statistical analysis will require some improvement before Acceptance. My major comments are to include a GLM analyses combining all predictor variables to investigate the relative degree as to which these host and environmental variables contribute to skin microbial variation, to include soil pH and body morphometric data as additional factors in analyses that could impact skin microbiome variation and to include analyses investigating how relative abundances of bacterial taxa are impacted by host and environmental factors. Further detailed comments on the previous issues and minor comments on statistical analysis are in the Additional Comments section. These comments shouldn’t discourage the authors because these issues can be easily addressed, and the manuscript is already in very good shape!

Validity of the findings

No comment

Additional comments

Abstract
Line 26-27: Here it is mentioned that host factors are important to skin microbial communities, but I think it’s important to add host environmental/extrinsic factors as well. Especially since the authors investigated the impacts of both intrinsic host and extrinsic environmental.
Line 40: Add “habitat use of different species” to ensure consistency with previous sentence.

Introduction
Line 48: Here I would just specify bacterial communities, instead of microbial, since this is what the manuscript is focused on and not any other fungal, archaea microbes.
Line 48-50: It would be good to have some additional reviews referenced here that are of particular interest to this study Jiménez & Sommer (2017: 10.2478/micsm-2013-0002) and Rebollar et al. (2020: doi.org/10.1655/0018-0831-76.2.167).
Line 62: Remove “However” as not opposing previous sentence so not necessary to write it.
Line 79-80: “through a survey of the skin microbiome and Bd infection status of amphibians” is redundant as it is said in the beginning of the sentence Bd infection, and the skin microbiome was assessed. I would just suggest considering removing this part.
Line 80-85: This is a bit confusing… was tadpole data collected from all Lithobates species, or is it just one couldn’t distinguish between the different species at the tadpole life stage?
Line 85-90: Expected results are really well stated, would be good to include what authors thought would be the expected results of correlation between Bd infection and microbiome.

Methods
Line 95-97: Thorough sampling in the area which is great! Especially dissections of subsites in site 1 is nice to see. Reference Table 1 here.
Line 99-100: Also make clear here that samples of bacterial communities were only taken from the year 2017. Or in line 103.
Line 100: It seems samples were only collected from two subsites the pool and stream, why then was site 1 divided into 5 subsites?
Lines 100 – 105: It’s not clear to me why only site 1 is subdivided into subsites, when all of the sites have similar subsites, i.e. pools and streams. Was this because site 1 is the only site where skin microbiome samples were taken?
Line 105: It would be good to know how these sites (and subsites) differ in vegetation and soil composition (if information is available of course). I think the information about how the waterbodies across sites varies is already described here, but it would be good to know how these sites differ in other ways as justification for why these subsites and sites would potentially produce differing or similar skin microbial communities.
Line 115: Did you take a sample of the dip nets to see how much skin microbes on the amphibians were influenced by the mode of capture?
Line 117: How would one avoid recaptures across days? Or was this something that one would prefer, i.e., potentially having recaps could see how stable microbiome composition is at sites across days but no way to statistically test this as amphibians weren’t marked?
Line 130: Great that you were able to rapidly store samples in cooler/more ideal conditions this is so difficult for many field studies! However, I wonder how long samples were stored at -20C and how this would impact microbiome data? Since we know ideal temperatures for bacterial DNA is -80C this could possibly impact results…
Line 133: Water and air temperature?
Line 134-135: What is the statistical test done to determine this? And is it possible to make this data accessible?
Line 139-140: Reference here Table 1.
Line 167: It seems that tadpole species could not be distinguished from each other, could the authors mention this earlier in the methods?
Line 198-199: See papers McMurdie & Homes (2014: https://doi.org/10.1371/journal.pcbi.1003531) and Gloor et al. (2017: https://doi.org/10.3389/fmicb.2017.02224) that show rarefaction of samples is a poor method to normalize data and one should include all sequencing data in the analyses. I do think it’s sensible to remove samples where alpha rarefaction curves have not plateaued since clearly the community wasn’t adequately sampled, but it doesn’t make sense to cut off sequences randomly from other samples. Additionally, it would be informative to see the alpha rarefaction curves, could this be included as a supplementary figure?
Line 215-216: Would be good to include UniFrac metrics as another measure of diversity since it looks at both presence/absence and phylogenetic distances?
Line 219: Not clear here whether two different PERMANOVAs were run, one looking at impact of host species and another looking at life stage. I assume it has to be two different PERMANOVAs since tadpoles could not be identified to species level…
Line 219: Why was site not included as a predictor variable in the beta diversity analyses?
Line 219: Specify levels of predictor variables amphibian species and life stage here.
Line 202: Why wasn’t soil pH included as a predictor for microbiome variation? Especially since there seems to be significant differences between subsites in pH which could lead to microbiome variation…
Line 202: Authors did collect body morphometric data from both adults and tadpoles. It would be interesting to include these measures in the analyses (through raw values and maybe a body condition metric if mass was also measured) and compare this to other studies where this is often done.
Line 202: A lot of Bd data was collected for amphibians in site 2 and 3, where does this data go in the current statistical analyses that just focuses on linking Bd data with microbiome data (which was not available for site 2 and 3). It seems like a lot of data was collected for Bd infection but not clear to me how this has been used in this manuscript… It would be interesting to also look at how the many environmental and host variables measured in this study predicts infection of amphibians with Bd. Even though this is a well-studied subject I think it’s still useful information authors can contribute to the literature!
Line 221-224: I think this sentence is misleading as it implies that betadisper results informs the variance among groups. Betadisper tests homogenous dispersion among groups – a condition/assumption of adonis. Whilst adonis tests for differences in variances among groups.
Line 222: Was betadisper analyses only done for host species and not life stage?
Line 230: Description of models aren’t very clear here. Were two models with different effects fitted and compared to the null model? Or were two different GLM analyses conducted for sampling sites and then for amphibian species and life stages? What does a full model look like? How many levels did each predictor variable have?
Line 230: Does the sampling sites variable distinguish between subsites?
Line 245: Would it be useful to have a GLM and PERMANOVA that investigates the relative degree variables contribute to variation in microbiome metrics (alpha and beta diversity)? I.e., a model including habitat, infection status, life stage etc. It makes sense to test whether there are microbiome differences in the different levels of the predictor variables investigated in this study. But I wonder if it would be useful to run an analysis to look at the relative contribution of each predictor variable to alpha and beta diversity microbiome variability. I understand the authors are limited in some cases, ex. it seems tadpoles could not be identified up to species level so life stage will not be included in this type of analysis, but surely infection status, site and soil pH (potentially?) could be included.

Results
Line 256: Relative abundances are discussed here, and I think the authors should include analyses to test for impact of host and environmental factors on bacterial abundances similarly to what they have done for alpha and beta diversity above.
Line 277: Are these statistics the additive of the two predictor variables? For example R2 of both amphibian species and life stages? This doesn’t make sense if it were so because tadpole life stages couldn’t be distinguished from the different species. It makes sense that there should be two PERMANOVA test results reported one for amphibian species and another for life stages for each of the beta diversity metrics…
Line 284-286: It would be good to make a comment here on what a significant dispersion result means for the adonis results. A quick look from the NMDS plot shows that L. palustris samples overlap with other species samples but show greater dispersion within species. This might mean that L. palustris microbiome does not vary from other amphibian species samples or tadpole samples.
Line 304-305: Not necessary to report permutest results, since there are no significant adonis results don’t need to test whether assumptions of adonis is met through permutest.
Line 307: In the methods it isn’t outlined that subsites underwent statistical analyses…
Line 308-310: I’m confused here as it seems that there were more samples taken from different subsites, but in the methods, it is stated samples were only taken from the stream and pool (two subsites).
Line 311: Why was there no tests for differences in beta diversity between subsites?

Discussion
Line 316-318: Just out of curiosity why would one expect to find low infection prevalence and the infection to be uniform across species?
Line 318-321: If infection prevalence is discussed here there should be some statistical tests and methods accompanying this. Currently the methods and results are focused on testing microbiome differences and linking microbiome and infection status, but there is no or little information on infection variation across species, habitat, years etc.
Line 325: Rather than “there were only” write “this study only had”
Line 326-328: Testing for differences in Bd infection among sites, species etc. should be outlined in the methods and results.
Line 331-336: There is a lot of discussion here on how come there is a non-significant result and relating this to sample sizes, mostly problems with experimental design etc. But it would be interesting to speculate on whether the results seen in this study is actually just that: infection of Bd doesn’t influence gut microbiome in this species at this site. Could it be that other habitat factors or host factors have a stronger influence then on the microbiome? Is there something that overcomes the relationship between Bd and the skin microbiome in this particular study? What do other studies that find similar results say?

Figure 1
Would be good to have an indication of sample sizes.
Map isn’t very high quality, can’t read all indications on the map.

Figure 2
It seems that relative abundance of families was different across species, but this has not been tested statistically.

Figure 4
Can it be indicated on the graphs where differences were significant between groups?

---

## Round 0.2 · accepted · Accept

I appreciate your taking into account reviewers' comments and hope that you found them relevant to improve your work.

Reviewer 1 ·

Basic reporting

Appreciation to all the authors for making the additional improvements and analysis. I am very happy to see my concerns/suggestions have been fully addressed.

I would recommend this manuscript for publication at PeerJ.

Experimental design

no comment.

Validity of the findings

no comment.

Additional comments

no comment.